# Luciferase Immunosorbent Assay Based on Multiple E Antigens for the Detection of Chikungunya Virus-Specific IgG Antibodies

Xiaoxia Li,[a] Xuan Wan,[b] Jinyue Liu,[a] Haiying Wang,[c] Anan Li,[d] Changwen Ke,[d] (ID) Shixing Tang,[c] Wei Zhao,[a] Shaoxi Cai,[b] (ID) Chengsong Wan[a]

[a]BSL-3 Laboratory, Guangdong Provincial Key Laboratory of Tropical Disease Research, School of Public Health, Southern Medical University, Guangzhou, China
[b]Department of Respiratory and Critical Care Medicine, Nanfang Hospital, Southern Medical University, Guangzhou, Guangdong, China
[c]Department of Epidemiology, School of Public Health, Southern Medical University, Guangzhou, China
[d]Guangdong Center for Disease Control and Prevention, Guangzhou, China

Xiaoxia Lia and Xuan Wan contributed equally to this article. Author order was determined by the corresponding author after negotiation.

**ABSTRACT** In recent years, the chikungunya virus (CHIKV) has continued to spread from local epidemics to nonnative habitats until eventually reaching pandemic status. Nonendemic areas such as China have also emerged as potential epidemic areas of CHIKV. Serological detection of CHIKV is the key to diagnosing and controlling the prevalence of this virus. In this study, we review the progress of the serological detection of the envelope (E) protein in CHIKV, and we provide a novel research assay and ideas for the serological detection of CHIKV. The luciferase immunosorbent assay (LISA) does not require species-specific labeled secondary antibodies for detection, making it universally suitable for tracking samples from various animals or carriers. At present, most research on CHIKV antigen detection technology tends to combine two or more proteins to avoid the decrease in detection ability caused by antigen mutation. Our results indicate that two or more kinds of CHIKV E antigens combined with LISA detection can improve the detection rate of anti-CHIKV immunoglobulin G (IgG) antibodies in CHIKV-infected patient sera and detect antibodies in the early stage of infection accurately and sensitively. After 235 days of infection, the anti-CHIKV IgG antibodies could still be detected in CHIKV-infected patients. All serum samples were tested with a detection rate of 100% after combining various recombinant CHIKV E antigens. Our proposed CHIKV-specific LISA could be a useful tool for serum diagnosis of CHIKV infection and serum epidemic research in areas where CHIKV is endemic, which would help to manage potential epidemics in the future.

**IMPORTANCE** At present, chikungunya virus (CHIKV) is still circulating in some parts of the world, and mutated strains have emerged, making it easier for the virus to spread among humans. With the continuous variation of CHIKV, its antigen variation leads to the decline of detection ability. In addition, the risk of transmission of CHIKV in areas where CHIKV is not endemic, such as China, has increased dramatically, which compels us to enhance the detection capacity of CHIKV and continuously monitor CHIKV antibody levels in the population. Real-time quantitative PCR (RT-PCR) detection technology will not be reliable when the infection time is chronic or in subclinical infection due to decreases in virus concentration, and an antibody detection technology must be adopted. In this study, multiple CHIKV envelope (E) antigens were used to detect anti-CHIKV IgG antibodies in serum for the first time. The new assay is characterized by convenient operation, high detection rate, and high sensitivity and has significance for early warning and monitoring. Moreover, it contributes to the prevention and control of CHIKV.

**KEYWORDS** chikungunya virus (CHIKV), envelope (E) antigens, luciferase immunosorbent assay (LISA), CHIKV antibodies, envelope protein

Address correspondence to Wei Zhao, 791478771@qq.com, or Chengsong Wan, gzwcs@smu.edu.cn.

The authors declare no conflict of interest.

The chikungunya virus (CHIKV) is an *Aedes* mosquito-transmitted reemerging *Alphavirus* that causes chikungunya fever, a debilitating disease (1). Since its reemergence of epidemic proportions in several southeast Asian countries during 2006 to 2010 (2, 3), CHIKV (namely, the Asian and Indian Ocean Lineage type [IOL] strains) has spread to Europe and American countries, where the virus was not previously observed (4). Infection by CHIKV typically results in mild and self-limiting disease in infected humans with a low fatality rate (~0.1%). However, acute and chronic disability with CHIKV infection leads to considerable impacts, including significant impacts on the quality of life for infected patients and considerable community and economic consequences (5–7).

Since the first outbreak of CHIKV in 1952, the virus has been prevalent in Central Africa, South Africa, West Africa, and Southeast Asia. CHIKV is prone to mutation, and the current types include the Asian type, East/Central/South African type (ECSA), West African type, and the IOL (8). Sequencing and evolutionary analysis of isolates from CHIKV outbreaks in India and the islands of the Indian Ocean from 2005 to 2007 showed that ECSA genotypes aggregated and evolved into the IOL genotypes, which adapted to the new vector *Aedes albopictus* through amino acid mutations in the encapsulation glycoproteins E1 and E2 (9, 10). The new E1-A226V mutation enhances the replication and transmission of CHIKV in *A. albopictus*. In 2010, an outbreak of CHIKV infection with the E1-A226V mutation occurred in the Guangdong Province, China (11). Therefore, rapid and convenient detection modalities for CHIKV are key to the comprehensive prevention and control of CHIKV. At the acute stage 4 to 6 days after onset, the antibodies detected in sera of CHIKV-infected persons are immunoglobulin M (IgM). Immunoglobulin G (IgG) antibodies can be detected in the serum 6 to 7 days after symptoms appear, and the viral RNA tends to disappear rapidly over time (4). IgG antibodies begin to appear approximately 1 week after symptoms and can last for several years. Antibody detection has several advantages, including a low price and on-site detection, and serological diagnosis is the most common detection assay at present. However, antibody detection also suffers from cross-reactivity issues with other *Alphaviruses*, cannot distinguish between recent past and acute infections, and its sensitivity varies according to the clinical setting (12). Indeed, when used to diagnose acute CHIKV infection, the serological sensitivity was only 4% to 22%, increasing to more than 80% after 1 week (13). At present, there is no gold standard immunological detection assay for CHIKV infection, and a standard clinical serological assay is urgently required.

CHIKV protective adaptive immunity is mainly provided by specific antibodies, especially those against E1 and E2 antigen epitopes (14). Receptor binding is the first step for viruses to enter cells. The MXRA8 receptor can bind to the CHIKV E2-E1 heterodimer on the surface of the virion, which contains two uniquely connected oriented chain exchange Ig-like domains. (15). CHIKV surface envelope proteins (E2 and E1) function as receptor-binding and fusion proteins, respectively. E1 and E2 are important proteins that participate in CHIKV replication and virus-host interactions (16). CHIKV E1-E2 is largely specific to the virus and was therefore used to develop and detect the anti-CHIKV antibodies (17, 18).

At present, an enzyme-linked immunosorbent assay (ELISA) based on CHIKV E antigen has been developed and applied (19). However, despite the high specificity of this ELISA for detection of anti-CHIKV IgG, it has low sensitivity and cannot be used for species-specific labeled antibodies in nonhuman samples. In this study, we developed a novel assay called luciferase immunosorbent assay (LISA) that can be used for the ultrasensitive detection of CHIKV infection with CHIKV E protein as antigen. Serum samples from patients infected with CHIKV, possibly cross-reactive sera from patients infected with dengue virus (DENV) and hepatitis C virus (HCV), and control samples that were validated as normal human sera were used to assess the detection capability of CHIKV LISA.

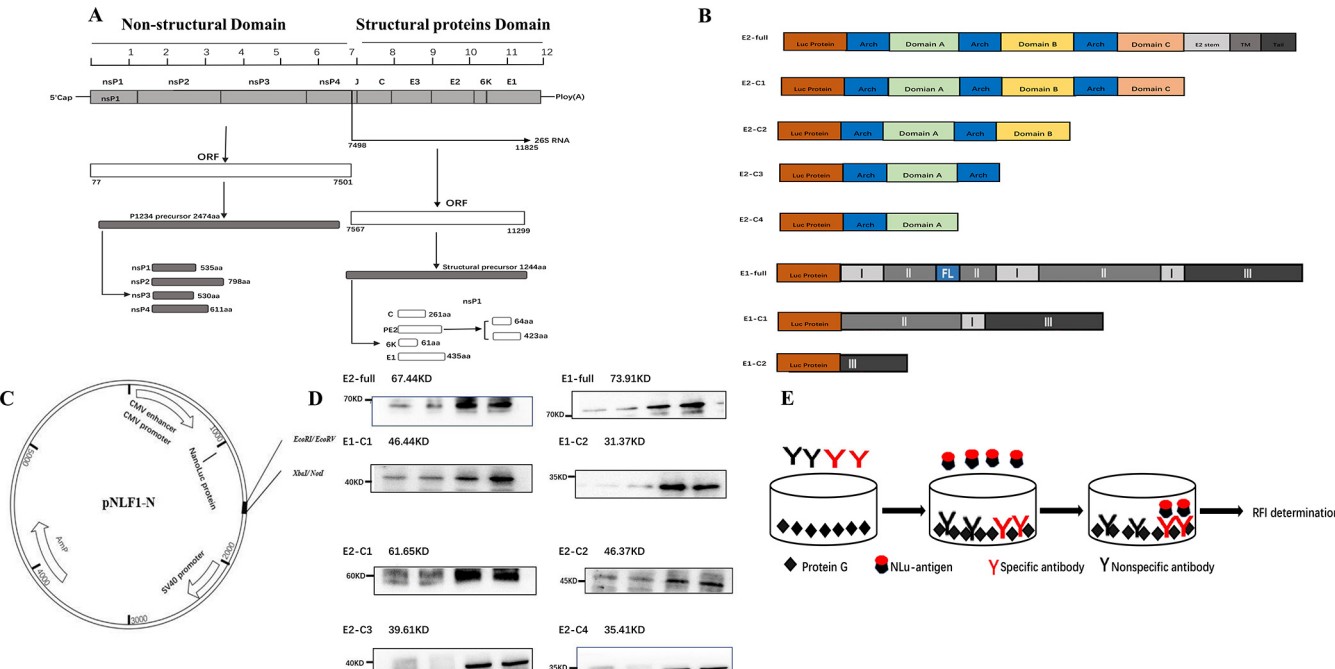

**FIG 1** (A) A brief discourse on CHIKV structure. (B) Cloning strategy of recombinant CHIKV E glycoprotein in pNLF1-N. Schematic diagram showing the chimeras used in sero-neutralization. Cloning strategy of recombinant CHIKV E2 glycoprotein in pNLF1-N. Schematic representation of the E2 and E1 glycoprotein, domain A, domain B, domain C, $\beta$-ribbon, transmembrane region (TM), and cytoplasmic tail (C tail). (C) A vector map of pNLF1-N with EcoRI/ EcoRV and NotI/ XbaI restriction sites. (D) Mouse anti-His ($\alpha$-His, at 1:7,000 dilution) are used to detect recombinant protein. (E) Schematic of the LISA and its validation based on different recombinant proteins.

## RESULTS

**Expression of CHIKV E antigens fused with Nano-luciferase.** Chikungunya virus (CHIKV) is a positive-sense RNA virus belonging to the *Togaviridae Alphavirus* family, which is transmitted to humans by *Aedes* mosquitoes. The 11.8-kbp genome was divided into two coding regions, each containing an open reading frame, encoding a total of nine proteins, which were divided into five structural proteins (C, E3, E2, 6K, and E1) and four nonstructural proteins (NSP1 to NSP4). It encodes a non-structural polymeric protein consisting of 2474 amino acids and a structural polymeric protein consisting of 1244 amino acids (Fig. 1A). E1 is a type II fusion protein, which mediates the fusion of virus envelope and cell membrane through the fusion peptide. E2 mediates the binding of receptor and attachment factor on the cell membrane and is the main target of neutralizing antibody. The N-terminal domain of E3 is the uncleared lead peptide of E2 and may help protect E1 fusion peptide (20). CHIKV protective adaptive immunity is mainly provided by specific antibodies, especially those against E1 and E2 antigen epitopes. The dominant linear epitope on the E2 glycoprotein is an important target for IgG neutralization in patients with early CHIKV infection. For the CHIKV E2 antigen and the receptor-binding domain (RBD) region, various fragments of CHIKV E1 and E2 antigens were used to establish a new approach for CHIKV detection. Standard molecular cloning was performed to clone the E1 glycoprotein (rE1: amino acids 1 to 499) and E2 glycoprotein (rE2: amino acids 1 to 425) genes (Fig. 1B) into a pNLF1-N (Promega, USA) vector (Fig. 1C) that contains a Nano-luciferase expression gene. In addition, His tags were added to the end of the E1 and E2 sequences; the purpose was to detect whether the protein is expressed normally. Eight Nano-luciferase expression plasmids were constructed, including full-length E1, E1-C1, E1-C2, E2 full-length, E2-C1, E2-C2, E2-C3, and E2-C4, all of which were transfected into 293T cells to express the luciferase fusion expressed antigen proteins (Fig. 1B).

Restriction endonuclease reactions and gel electrophoresis confirmed these plasmids. Finally, the recombinant protein was detected using the mouse monoclonal antibody against CHIKV E and His tag fusion expression protein by Western blotting

**TABLE 1** Sample information in this study

| Reference no. | Gender | Age | Date of onset (yr-mo-day) | Sampling date (year-mo-day) | Place of onset | CHIKV RT-PCR[a] |
|---|---|---|---|---|---|---|
| 1 | Male | 67 | 2010-10-10 | 2010 | Dongguan, China | + |
| 2 | Female | 65 | 2010-10-10 | 2010 | Dongguan, China | + |
| 3 | Female | 32 | 2010-10-14 | 2011-6-7 | Dongguan, China | + |
| 4 | Female | 65 | 2010-10-12 | 2010 | Dongguan, China | + |
| 5 | Male | 49 | 2010-10-6 | 2010 | Dongguan, China | + |
| 6 | Male | 56 | $-$[b] | 2011-6-7 | Dongguan, China | + |
| 7 | Female | 55 | 2010-10-7 | 2010 | Dongguan, China | + |
| 8 | Female | 25 | 2010-9-30 | 2010-10-1 | Dongguan, China | + |
| 9 | Female | 43 | 2010-10-4 | 2010-10-1 | Dongguan, China | + |
| 10 | Female | 47 | 2010-9-21 | 2010-10-1 | Dongguan, China | + |
| 11 | Male | 20 | 2010-9-30 | 2010-10-1 | Dongguan, China | + |
| 12 | Female | 75 | 2010-10-7 | 2010 | Dongguan, China | + |
| 13 | Male | 29 | 2010-10-1 | 2010-10-3 | Dongguan, China | + |
| 14 | Female | 39 | 2010-9-22 | 2010-9-24 | Dongguan, China | + |
| 15 | Female | 58 | 2010-9-26 | 2010-10-1 | Dongguan, China | + |
| 16 | Female | 62 | 2010-9-21 | 2010-9-24 | Dongguan, China | + |
| 17 | Male | 32 | 2010-10-7 | 2010 | Dongguan, China | + |
| 18 | Female | 55 | 2010-10-5 | 2010 | Dongguan, China | + |
| 19 | Male | 38 | 2010-10-2 | 2010 | Dongguan, China | + |
| 20 | Female | 68 | 2010-10-7 | 2010 | Dongguan, China | + |

[a]CHIKV RT-PCR test result is positive.
[b]–, data loss.

(Fig. 1D). After gene sequencing and Western blotting, the sequence comparison results confirmed the correct construction and expression of the CHIKV E plasmid. Therefore, LISA based on multiple E antigens could be applied to detect CHIKV-specific IgG (Fig. 1E).

**Anti-CHIKV IgG antibody detection by LISA.** The dilution ratio of CHIKV-infected patient sera was 1:1,000. The control samples were tested negative for nucleic acid and serum (commercial kit) IgM and IgG antibodies. There were 20 samples from patients diagnosed with CHIKV infection (reference numbers 1 to 20). Samples were collected from patients attending the Guangdong Center for Disease Control and Prevention (China) during the 2010 outbreak of CHIKV. These were acute samples collected from viremic patients between day 2 and day 11 after disease onset and one convalescent-phase serum collected 8 months later. In previous studies, real-time PCR targeting the E1 region was used to quantify the viral load and was combined with clinical symptoms to support a diagnosis of CHIKV infection (Table 1).

We comprehensively analyzed the relative fluorescence intensity (RFI) value of healthy people and the CHIKV-infected population. To ensure specificity, we compared the mean + standard error of the mean (SEM) in the CHIKV-infected population, which is approximately two times the RFI value of the control group. We defined the anti-CHIKV IgG antibody-positive RFI cutoff value as $5 \times 10$. When CHIKV-infected patients were tested with the LISA assay, a result was considered positive if the RFI value was higher than $5 \times 10^4$ (Fig. 2).

We constructed eight recombinant proteins based on the CHIKV E antigen. Serum samples from healthy subjects and patients with CHIKV were detected. We defined all samples with values higher than $5 \times 10^4$ as positive for CHIKV IgG antibodies. The sample numbers of positive results were as follows: E1-full (numbers 16 and 17), E1-C1 (numbers 16 and 17), E1-C2 (numbers 7, 16, and 17). E2 full-length (numbers 3, 5, 10, 19, and 20), E2-C1 (numbers 3, 10, 12, 13, 16, 17, 18, 19, and 20), E2-C2 (numbers 2, 3, 8, 10, 11, 12, 14, 16, 17, and 19), E2-C3 (numbers 1, 4, 5, 6, 7, 11, 15, 16, and 17), and E2-C4 (numbers 3, 6, 9, 11, 12, 14, 16, 17, 18, 19, and 20). The constructed LISA detection assay based on CHIKV E1 and CHIKV E2 antigens could detect anti-CHIKV IgG antibodies in some patient sera with CHIKV infection (Table 2). The recombinant Nano-luciferase CHIKV E protein antigens in the sera of patients showed that the recombinant CHIKV E1 protein could only detect CHIKV IgG antibodies in 2 to 3 of 20 CHIKV-infected samples, which was

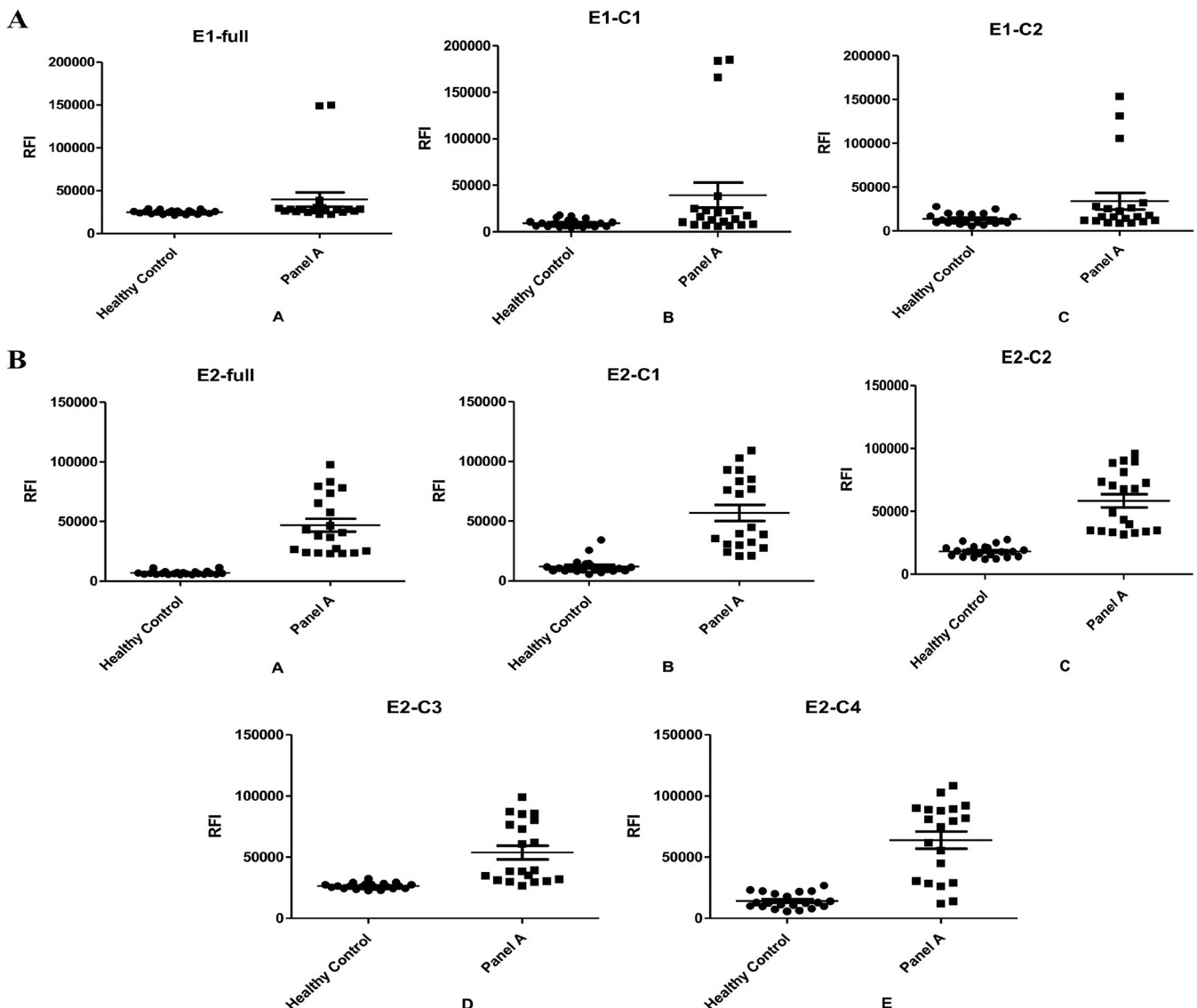

**FIG 2** Serum samples from healthy subjects ("healthy control") and patients with CHIKV ("panel A") were assessed for anti-CHIKV IgG antibodies by LISA. (A) LISA can detect samples with E1-full (numbers 16 and 17), E1-C1 (numbers 7, 16, and 17), and E1-C2 (numbers 7, 16, and 17). (B) LISA can detect samples with E2 full-length (numbers 1, 3, 5, 6, 10, and 19), E2-C1 (numbers 3, 10, 12, 13, 16, 17, 18, 19, and 20), E2-C2 (numbers 2, 3, 8, 10, 11, 12, 14, 16, 17, and 19), E2-C3 (numbers 1, 4, 5, 6, 7, 11, 15, 16, and 17), and E2-C4 (numbers 3, 6, 9, 11, 12, 14, 16, 17, 18, 19, and 20).

a result similar to those produced with the existing commercial kits. CHIKV E2 recombinant protein could detect more than half of patient samples. The assay could detect anti-CHIKV IgG antibodies in patient sera. Recombinant protein CHIKV E2 showed better results than recombinant antigen CHIKV E1 (Fig. 3A).

Due to the sample size limitation, the sample data did not conform to the normal distribution. Therefore, the Wilcoxon rank sum test was used to analyze the detection conditions based on CHIKV E1 and CHIKV E2, and the $P$ value of the two-tailed data test was 0.024 ($P < 0.05$), which was statistically significant (Fig. 3B).

**Optimal antigenic domain for anti-CHIKV IgG detection.** We next focused on the CHIKV E2-C1/CHIKV E2-C3/CHIKV E2-C4 LISA to further evaluate the detection ability of anti-CHIKV IgG. CHIKV E2-C4 could detect anti-CHIKV IgG antibodies in the sera of patients in the acute stage and could detect all the anti-CHIKV IgG antibodies in the sera of patients in the convalescent stage (about 8 months) (Fig. 3B). Anti-CHIKV IgG antibodies could not be detected in all samples using the existing commercial kits (only two were detected), owing to the long storage time. In contrast, our novel test

**TABLE 2** Construction of eight assays based on CHIKV E antigens to detect CHIKV-infected patient sera antibodies

|  | E1-full | E1-C1 | E1-C2 | E2-full | E2-C1 | E2-C2 | E2-C3 | E2-C4 |
|---|---|---|---|---|---|---|---|---|
| Anti-CHIKV IgG-negative | 1/2/3/4/5/6/7/8/ 9/10/11/12/ 13/14/15/18 | 1/2/3/4/5/6/7/ 8/9/10/11/ 12/13/14/ 15/18 | 2/3/4/5/6/7/8/ 9/10/11/12/ 13/14/15/ 18 | 1/2/4/6/78/9/ 11/12/13/ 14/15/16/ 17/18 | 1/2/4/5/6/7/8/ 9/11/14/15 | 1/4/5/6/7/9/ 13/15/18/ 20 | 2/3/7/8/9/10/ 12/13/14/ 18/19/20 | 1/2/4/5/7/8/ 10/13/15 |
| Anti-CHIKV IgG-positive | 16/17 | 7/16/17 | 1/16/17 | 1/3/5/6/10/19 | 3/10/12/1316/ 17/18/19/ 20 | 2/3/8/10/11/ 12/14/16/ 17/19 | 1/4/5/6/7/11/ 15/16/17 | 3/6/9/11/ 12/14/16/ 17/18/19/ 20 |
| Total | 2 | 3 | 3 | 6 | 9 | 10 | 9 | 11 |

was able to detect a higher positive rate with LISA E2-full CHIKV IgG (30%), LISA E2-C1 CHIKV IgG (45%), LISA E2-C2 CHIKV IgG (50%), LISA E2-C3 CHIKV IgG (45%), and LISA E2-C4 CHIKV IgG (55%). In the experimental process, CHIKV E2-C4 identified samples in the acute and recovery phases (Table 3). The results indicated that, to a certain extent, there are more anti-CHIKV IgG antibody-binding sites in Arch1 and domain A, and these sites may be hidden in the E2-full antigen protein due to spatial folding or other factors. In addition, E2-C2 had a higher positivity rate than E2-C1 and E2-C3. This suggests that there are also CHIKV IgG antibody-binding sites in domain A and domain B. However, we believe that domain A has a more significant effect than domain B.

An important target for antibody binding was found in the Arch1 and A domains in the CHIKV E2 antigen. The dominant linear epitope on the E2 glycoprotein is a target for IgG neutralization in patients with early CHIKV infection. However, our results also confirmed that the E2-full antigen could not detect IgG antibodies better in the sera of patients with CHIKV. It may be that the RBD, which is important for binding to antibodies, is hidden after protein folding. As a result, not all the IgG antibodies in the sample could bind to the CHIKV E2-full antigen.

The recombinant CHIKV E2 antigen was used to detect IgG in the sera of CHIKV-infected patients, and different E2 recombinant antigens could be used to detect different samples. Compared with the use of individual antigens, after binding all recombinant E2 antigens, we found that the CHIKV LISA assay detected more anti-CHIKV IgG antibody samples, with a specificity of 100% (Table 3).

**Sensitivity and cross-reactivity of the LISA assay.** RFI was performed using two commercial kits and LISA-positive samples. The maximum dilution ratio of positive samples was 1:25,600, and the lowest concentration detection was 0.235 ng/mL. The detection result of CHIKV E2 antigen was better than that of E1 antigen (Fig. 3B), the LISA assay based on 5 fragments (full, C1, C2, C3, and C4) of E2 antigen was further used for CHIKV-infected human sera. The assay is well described to characterize the binding domains of CHIKV IgG antibodies. The well-performing E2-C4 sequence failed to detect IgG antibodies in positive samples at a dilution of 1:25,600, while E2-C1, E2-C2, and E2-C3 LISA could not make this distinction at a dilution of 1:5,000 to 1:6,400. Thus, the sensitivity of the E2-C1, E2-C2, and E2-C3 LISA was at least 4-fold lower than that of the E2-C4 LISA (Fig. 4A). The ELISA for detecting CHIKV-infected IgG antibody-positive sera dilution range was 50 to 200 times, and the sample dilution range for LISA detection was 1,000 to 5,000 times. LISA could not detect CHIKV IgG-positive samples after about a 5,000 times dilution, and the sensitivity of the LISA was about 25-fold higher than that of the ELISA (Fig. 4B).

The within-run standard deviation and coefficient of variation values were then calculated. The three groups included the control group, the anti-CHIKV infection IgG antibody-negative group, and the anti-CHIKV infection IgG antibody-positive group. Three samples were measured in each group, and each sample was tested three times. The degrees of variation of nine samples were 6%, 7%, 6%, 4%, 0.7%, 1.5%, 0.78%, 4.1%, and 9.3%, and the coefficient of variation of each group was <10% (Fig. 4C).

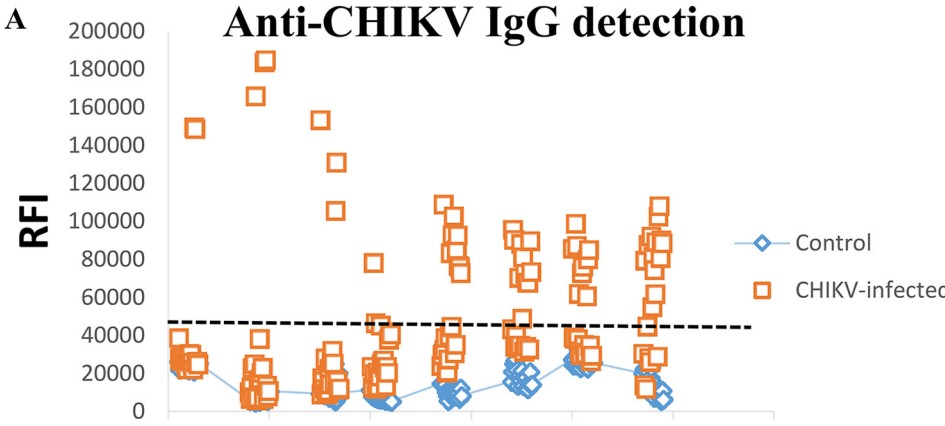

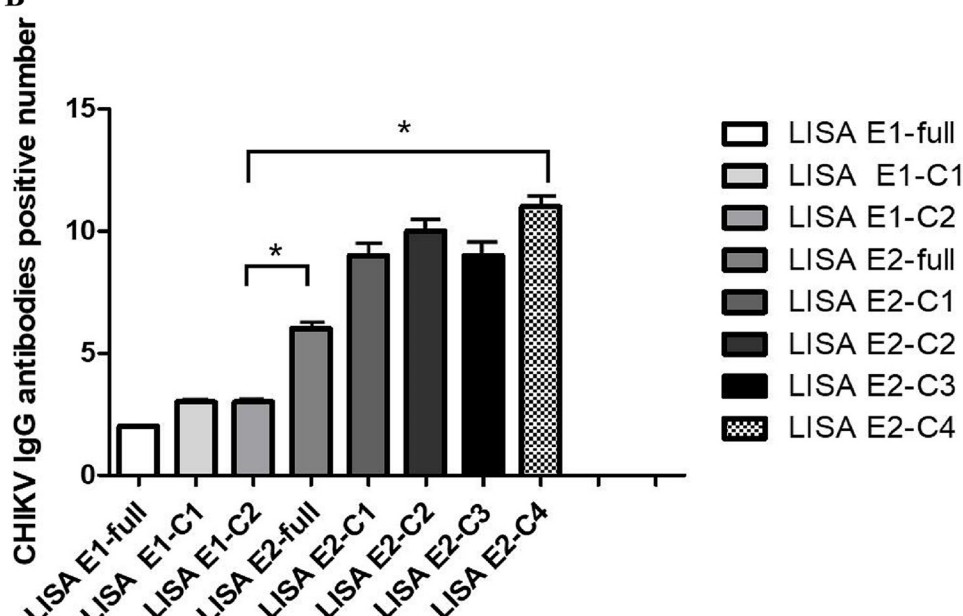

FIG 3 (A) We defined the anti-CHIKV IgG antibody-positive RFI cutoff value as $5 \times 10^4$ and consider it a positive result when CHIKV-infected patients were tested with the LISA assay. The eight assays could detect anti-CHIKV IgG antibodies in patient sera by LISA; serum data were collected. (B) Comparison of CHIKV E1 antigen and CHIKV E2 antigen detection sample volume. The C2 antigen group with the largest sample size detected by CHIKV E1, the E2-full antigen group with the lowest sample size detected by CHIKV-E2, and the C2-C4 antigen group with the largest sample size detected by CHIKV E2 were statistically analyzed. Data that are statistically significant are marked with an asterisk; *, $P < 0.05$.

We also explored the cross-reactivity of the CHIKV E-based LISA using a total of 47 convalescent-phase serum samples from people infected with different viruses of the *Flaviviridae* family, including DENV and HCV (Table 4). Analysis of variance and two-sample *t* test showed that there was no significant difference ($P > 0.1$) (Fig. 4D). Furthermore, the LISA method based on CHIKV E antigens did not detect IgG antibodies from the sera of patients with dengue fever and hepatitis C (Table 3).

**Comparison of the novel LISA with the commercial ELISA for detecting anti-CHIKV IgG and IgM.** We next focused on the ability of CHIKV E2-C4 recombinant antigen carrying Nano-luciferase to detect anti-CHIKV IgG antibodies in CHIKV-infected sera. In the ELISA kit testing, our results showed that only two samples could detect anti-CHIKV IgM and IgG antibodies in CHIKV-infected sera (Fig. 5A). Owing to the long storage time of the samples (approximately 10 years), the detection rate of positive

**TABLE 3** The results of cross-reactivity

| Analysis of detection assays | CHIKV serum ($n = 20$) specificity (%) | DENV serum ($n = 40$) specificity (%)[a] | HCV serum ($n = 5$) specificity (%)[a] | Control ($n = 20$)[a] |
|---|---|---|---|---|
| LISA E1-full CHIKV IgG | 2 (10%) | – | – | 0 |
| LISA E1-C1 CHIKV IgG | 3 (15%) | – | – | 0 |
| LISA E1-C2 CHIKV IgG | 3 (15%) | – | – | 0 |
| LISA E2-full CHIKV IgG | 6 (30%) | – | – | 0 |
| LISA E2-C1 CHIKV IgG | 9 (45%) | – | – | 0 |
| LISA E2-C2 CHIKV IgG | 10 (50%) | – | – | 0 |
| LISA E2-C3 CHIKV IgG | 9 (45%) | – | – | 0 |
| LISA E2-C4 CHIKV IgG | 11 (55%) | 0 | 0 | 0 |
| LISA multiple E antigen combination IgG | 20 (100%) | 0 | 0 | 0 |
| ELISA CHIKV IgG | 2 (10%) | – | – | 0 |
| ELISA DENV NS1 IgG | | 10 (25%) | – | 0 |
| ELISA HCV IgG | | | 4 (100%) | 0 |

[a]0, no positive results were detected; –, no relevant data results were made.

samples of anti-CHIKV IgG antibodies was low. The optimized LISA anti-CHIKV detection method could detect more IgG antibody-positive samples in CHIKV-infected sera.

We used ELISA E1 and LISA E2-C4 detection assays to detect the number of CHIKV IgG antibody-positive samples, and, according to a two-sample rate chi-square test, the $P$ of the two-tailed data test was 0.004 ($P < 0.05$), which was statistically significant (Fig. 5B). Our best antigen based on the CHIKV E antigen could detect approximately 50% of the IgG antibody-positive samples. After 180 days of infection, the anti-CHIKV IgG antibodies could still be detected in patients. All anti-CHIKV IgG antibody serum samples were detected with a detection rate of 100% after the combination of various CHIKV E2 antigens in CHIKV-infected patient sera, including all the recombinant CHIKV E2 antigens; this suggests that the combination of two or more antigenic proteins will increase the detection rate of anti-CHIKV IgG antibodies in patients with CHIKV infection.

## DISCUSSION

In October 2010, a small endemic outbreak of CHIKV occurred in a village in Guangdong Province, southern China (21). Three phylogenetically distinct groups of CHIKV with distinct antigenic properties have been identified: the Asian genotype, the West African genotype, and the East/Central/South/African (ECSA) genotype. Notably, the single-amino acid mutation (A226V) in the E1 protein that determined infectivity and vector specificity was present in this Chinese strain. However, the native and imported strains showed different sequence variations. The E1 gene *A226V* and E2 gene *V264A* were detected in the three imported strains, and the unique substitution of the E1 gene *S250P* and E2 gene *H313Y* could be observed in four of the imported strains (11, 22). These significant variations may be responsible for some outbreaks. There may be some mutants in the existing samples, but our kit can detect the anti-CHIKV IgG antibodies of all samples preserved so far.

Currently, most CHIKV antigen detection technologies tend to combine two or more antigens to avoid a decrease in detection ability caused by antigen mutations (16, 23–25). Chua demonstrated the neutralization effect of IgM at different times after infection and examined the independent role of IgM and IgG in the neutralization ability of human immune sera in the early stages of infection, including differences in neutralization epitopes. IgM can detect CHIKV on day 4 of symptom onset, earlier than IgG. IgM preferentially binds and targets E1-E2 glycoprotein epitopes on the surface of CHIKV rather than individual E1 or E2 (26). Monoclonal antibodies induced by Tuekprakhon using CHIKV C protein and E1 protein are superior to 6 K and E1 proteins. The induced monoclonal antibodies show various reactivities against the current globally prevalent genotypes (27). The envelope (E) protein genes of CHIKV E1 and CHIKV E2 were subcloned into baculovirus vector pFAST-HT to obtain the plasmid pFAST-HT-CHIKV-E1-E2 of the recombinant protein, which showed that recombinant CHIKV E1

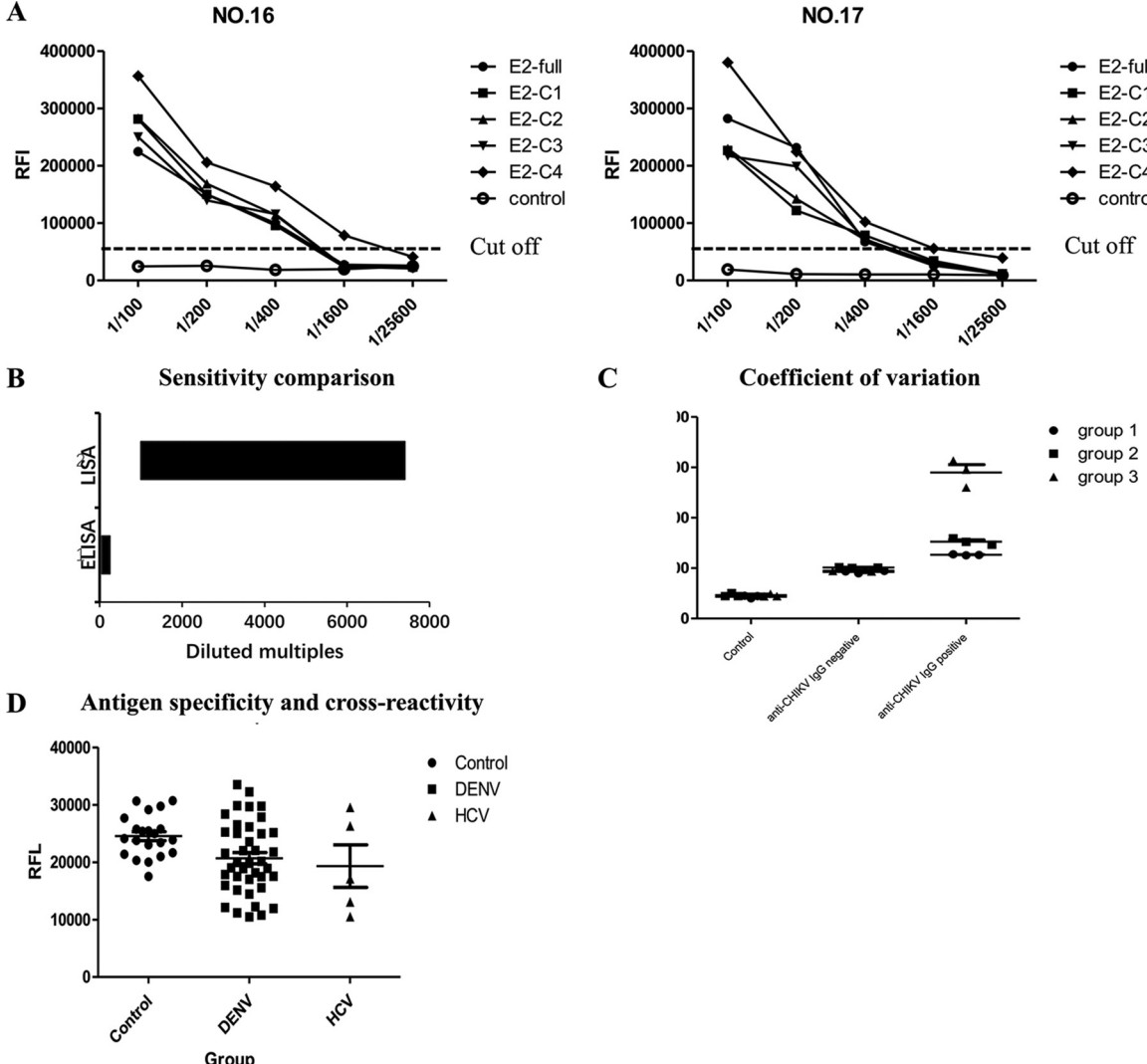

**FIG 4** (A) Serum samples from a patient with CHIKV infection, collected on day 3 (number 16) and day 235 (number 17) after admission in the hospital, were serially diluted, and the relative fluorescence intensity (RFI) was determined using the method as described above. The cutoff value was decided based on values from 21 healthy donors (numbers 66 to 87). (B) Sensitivity comparison between LISA and ELISA. (C) Coefficient of variation for LISA. (D) Antigen specificity and cross-reactivity ($P > 0.1$).

and CHIKV E2 envelope proteins have the potential of diagnostic reagents for serological diagnosis due to their high sensitivity and specificity (28).

Our results were consistent with those reported by Voss, in that the majority of the CHIKV E antigen located in the terminal CHIKV E2 antigen. The results show that Arch1 and A domains of CHIKV-E2 are important antibody-binding targets. The dominant linear epitope on CHIKV E2 glycoprotein is an important target for IgG neutralization in patients with early CHIKV infection. Our results also confirmed that CHIKV E2-full antigen could not detect all IgG antibodies in the sera of CHIKV-infected patients. This may be that the RBD is hidden after the protein is folded and expressed to form a spatial structure, but RBD is an important domain that binds to antibodies (29). Therefore, the IgG antibodies in the sample cannot completely bind to the CHIKV E2-full antigen. We compared our LISA results with those of the commercially available ELISA kits, which have been validated and used in many studies. These results will provide a new basis for developing an anti-CHIKV IgM/IgG antibody detection kit.

The preservation of clinical sera is relatively strict, and its antibody concentration

**TABLE 4** Summary of sample detection in this study

| Group | Reference no. | Source (human) Region | Source (human) Notes | CHIKV RT-PCR E1[a] | DENV RT-PCR[a] | ELISA (IgG)[a] | HCV RT-PCR[a] |
|---|---|---|---|---|---|---|---|
| CHIKV serum (n = 20) | 1, 2 | China | | + | − | − | |
| | 3 | China | About 8 months | + | − | − | |
| | 4–7 | China | | + | − | − | |
| | 8 | China | 3 days, acute phase | + | − | − | |
| | 9 | China | 3 days, acute phase | + | − | − | |
| | 10 | China | 11 days, acute | + | − | − | |
| | 11 | China | 2 days, acute phase | + | − | − | |
| | 12 | China | | + | − | − | |
| | 13 | China | 2 days, acute phase | + | − | − | |
| | 14 | China | 2 days, acute phase | + | − | − | |
| | 15 | China | 6 days, acute phase | + | − | − | |
| | 16 | China | 3 days, acute phase | + | − | + | |
| | 17 | China | | + | − | + | |
| | 18–20 | China | | + | − | − | |
| DENV serum (n = 40) | 21 | China | | − | + | − | |
| | 22 | China | | − | + | + | |
| | 23–33 | China | | − | + | − | |
| | 34–36 | China | | − | + | + | |
| | 37 | China | | − | + | − | |
| | 38 | China | | − | + | + | |
| | 39–41 | China | | − | + | − | |
| | 42 | China | | − | + | + | |
| | 43, 44 | China | | − | + | − | |
| | 45 | China | | − | + | + | |
| | 46–50 | China | | − | + | − | |
| | 51 | China | | − | + | + | |
| | 52 | China | | − | + | + | |
| | 53 | China | | − | + | − | |
| | 54 | China | | − | + | + | |
| | 55–60 | China | | − | + | − | |
| HCV serum (n = 5) | 61–64 | China | | − | − | + | + |
| | 65 | China | HIV + HCV coinfection | − | − | + | + |
| Control (n = 21) | 66–87 | China | | − | − | − | − |

[a]+, RT-PCR test result is positive; −, RT-PCR test result is negative.

will be greatly reduced after being stored at 2 to 8°C for 1 week. In practice, the application of ELISA for quantitative detection should be completed within 1 week as far as possible but at most within 1 month. The positive serum results can be kept stable at −20°C for 6 months for qualitative detection. Improper storage of sera or long-term storage leads to degradation of antibodies. Because of these problems, further development of highly sensitive and specific detection assays is needed (30–32).

As serum samples are often stored long term, IgG degradation is common, which results in interference in sample detection. However, LISA could detect anti-CHIKV IgG at a lower limit of detection (LOD) than ELISA. Traditional serological assays require purification of monoclonal antibodies or antigens for detection, which is usually a complex process. ELISAs require species-specific enzyme-conjugated antibodies, which are not necessary for LISA. LISA can be used to detect antibodies to emerging infectious diseases quickly and conveniently by direct cleavage of proteins after fusion expression without antigen purification and expression modification. LISA has high sensitivity and can be used for quantitative detection, which has a wide range. Moreover, the application of ELISA is limited because it is difficult to obtain generic-specific antibodies from vectors (e.g., mites and mosquitoes) and nonlaboratory animals (e.g., bats), while LISA does not require species-specific labeled secondary antibodies for detection. The LISA assay can be used for animal tracing research and epidemiological survey, which will provide support for early surveillance and warning of emerging infectious diseases. In conclusion, the LISA detection assay based on multiple CHIKV E antigens can detect the anti-CHIKV IgG antibodies in the sera of patients with early CHIKV infection

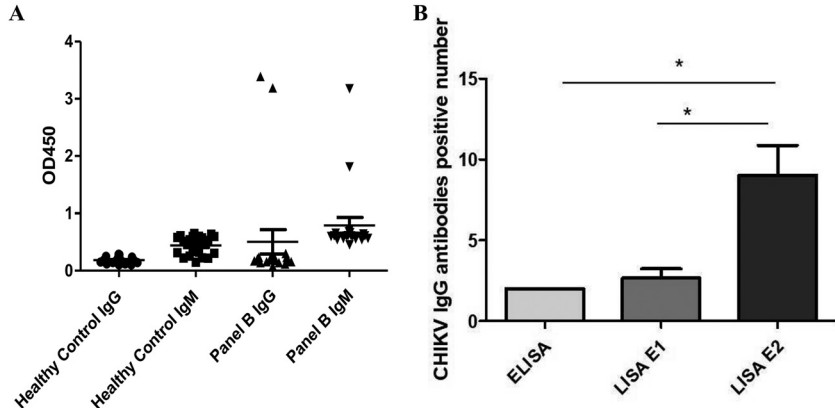

**FIG 5** (A) ELISA detection of anti-CHIKV IgM and IgG antibodies in CHIKV-infected sera. (B) ELISA and LISA E2 detection assays to detect the number of CHIKV IgG antibody samples. CHIKV IgG antibody was detected by LISA E2 and LISA E2. Data that are statistically significant are marked with an asterisk; *, $P < 0.05$.

accurately and sensitively. We describe a novel assay for detecting specific CHIKV IgG antibodies that can be a useful tool for the serodiagnosis of CHIKV infection and sero-prevalence studies in regions of endemicity; this novel technology will contribute to the management of potential future epidemics.

## MATERIALS AND METHODS

**Immune serum panels.** In September 2010, a patient with symptoms of CHIKV was reported at a local community clinic on the outskirts of Dongguan, Guangdong Province. The first patient became ill on 1 September, and the rapid increase in cases of CHIKV fever after 19 September suggests an outbreak of CHIKV infection in the area. The outbreak lasted about 60 days, peaking in late September/early October. There were no deaths or severe cases in this outbreak, and most patients recovered within a week of symptom onset. However, some of the older people reported joint pain 2 weeks later. Nucleic acid, antibodies, and virus were detected in the sera of patients infected with CHIKV. Real-time quantitative PCR (RT-PCR) was used to analyze the DNA sequence of the amplified CHIKV envelope 1 (E1) to infer the possible source of transmission. Genetic analysis of the 325-nucleotide (nt) fragment of the E1 gene obtained in this study showed that all seven sequences clustered in a unique branch within the Indian Ocean clade of the ECSA genotype and close to Thailand (GQ870312, FJ882911, GU301781), Malaysia (FJ998173), Taiwan (FJ807895), and China (GU199352, GU199353) isolates (98% to 99%).

All the serum samples were collected from patients during the 2010 CHIKV outbreak in Guangdong Province. After the patients were diagnosed with CHIKV infection, patient blood was drawn, and the supernatants were centrifuged after standing. The collected patient sera were stored at −80°C.

This study used four panels of serum samples. Panel A was composed of 20 (reference numbers 1 to 20) samples collected from patients attending Guangdong Center for Disease Control and Prevention (China) during the 2010 outbreak of CHIKV. These were acute samples collected from viremic patients between day 2 and day 11 after disease onset and one convalescent-phase serum sample collected 8 months later. Viral loads were quantified using real-time PCR targeting the E1 region in a previous study. Patient sera IgG and IgM antibodies were also detected (Table 1). Panel B was composed of 40 (reference numbers 21 to 60) samples collected from patients attending the Guangdong Center for Disease Control and Prevention (China). Samples with consistent clinical symptoms and RT-PCR and antibody detection were diagnosed with DENV infection. Panel C was composed of five (reference numbers 61 to 65) samples collected from patients attending the Central Laboratory of Liver Diseases, Nan Fang Hospital, Southern Medical University, Guangdong, China. Panel D was composed of 21 samples (reference numbers 66 to 87) from 21 healthy controls with no past infection of CHIKV, confirmed by the absence of CHIKV IgG antibodies by serum neutralization assay. We also explored the cross-reactivity of CHIKV E-based LISA using a total of 45 convalescent-phase serum samples from people infected with different viruses of the *Flaviviridae* family, including DENV and HCV (Table 4).

**Cell culture and transfection.** Cryopreservation tubes containing HEK 293T cells were removed from liquid nitrogen and immersed directly in a water bath at a constant temperature of 37°C, without shaking, to promote rapid thawing of the cells. The 293T cells were resuscitated in Dulbecco's Modified Eagle's Medium (DMEM; Gibco, New Zealand, C11995500BT-2) containing 10% fetal bovine serum (Gibco, New Zealand, 10270-106), and the cell culture flask was placed in an incubator at 37°C. The luciferase recombinant plasmid was transfected into 293T cells with Lipofectamine 3000 transfection reagent (Invitrogen, USA, L3000-015) to express luciferase target antigen. After the reaction system was

**TABLE 5** Recombinant antigen primer DNA sequence

| Amplified gene | Primer name[a] | Primer sequence | Fragment size (amino acids) | Restriction site |
|---|---|---|---|---|
| E2-full | F-R | GAATTCATGCGCAGCACCAAGGACAACTTC | 431 | EcoRI |
| | F-F | TCTAGATTAATGGTGATGGTGATGATGGGCCGCTTTAGCTGTTCTGATG | | XbaI |
| E2-C1 | C1-R | CCGGAATTCATGCGCAGCACCAAGGACAACTTC | 370 | EcoRI |
| | C-F | TGCTCTAGATTAATGGTGATGGTGATGATGGTATAGCTCATAATAATACAG | | XbaI |
| E2-C2 | C2-R | CCGGAATTCATGCGCAGCACCAAGGACAACTTC | 244 | EcoRI |
| | C2-F | TGCTCTAGATTAATGGTGATGGTGATGATGATACTGCCACTTTTTGTGATT | | XbaI |
| E2-C3 | C3-R | CCGGAATTCATGCGCAGCACCAAGGACAACTTC | 179 | EcoRI |
| | C3-F | TGCTCTAGATTAATGGTGATGGTGATGATGGGGCATGTGTACCTCTATCTC | | XbaI |
| E2-C4 | C4-R | CCGGAATTCATGCGCAGCACCAAGGACAACTTC | 146 | EcoRI |
| | C4-F | TGCTCTAGATTAATGGTGATGGTGATGATGTTCCCGGCCTATCACAGGAGG | | XbaI |
| E1-Full | F-R | CATCATCACCATCACCATATCAGAACA | 505 | EcoRV |
| | F-F | AAGCACCACGATTAGAATCAG | | NotI |
| E1-C1 | C1-R | GAATTCATGGCCACCTACCCATTCATGTGGGGCGCCT | 244 | EcoRV |
| | C1-F | TCTAGAATGGTGATGGTGATGATGCACACTTGCCTTTCTTGCTGG | | NotI |
| E1-C2 | C2-R | GAATTCATGGATTACAAGGATGACGACGATAAGGCCACCGTGCACGTGCCG | 100 | EcoRV |
| | C2-F | TACTCTTCTAGACACACTTGCCTTTCTTGCTGG | | NotI |

[a]F, forward; R, reverse.

prepared, it was left still for 10 to 15 min. The reaction medium was transferred into 293T cells after changing the medium, and the cells were cultured in a $CO_2$ incubator at 37°C for 48 h before being collected.

**Recombinant luciferase fusion protein.** CHIKV E1 and E2 DNA sequence fragments were inserted into the pNLF1-N (Promega, USA, N1351) plasmid, and the recombinant plasmid was transfected into 293T cells to express the recombinant protein. The recombinant antigen sequence of CHIKV rE2 (strain S27-African prototype) is 1,275 nt in length, without the transmembrane region (GenBank accession number AF369024). Standard molecular cloning was performed to clone the E1 glycoprotein (rE1; amino acids 1 to 499) and E2 glycoprotein (rE2; amino acids 1 to 425) open reading frames into a pNLF1-N vector. According to the structural characteristics of E2, we designed five amino acid sequences: a complete E2 glycoprotein, an E2 glycoprotein (rE2, from amino acids 1 to 362) without the transmembrane and cytoplasmic tail regions, and three recombinant proteins based on E2 antigen (Fig. 1B). The five amino acid sequences were designed to distinguish between E2 antigen-specific antibody-binding sites and extensive antibody-binding sites. Three amino acid sequences were designed to distinguish E1 antigen-specific antibody-binding sites. Luciferase fusion of E1 and rE2 was performed using a short linker GGGS-His (8×)-GGGG (Table 5).

**Western blotting.** The cell lysate proteins were resolved with 10% SDS-PAGE under nonreducing or reducing conditions and electrotransferred onto a nitrocellulose (NC) membrane (China). The membrane was blocked with 5% skim milk in 0.05% Tris-buffered saline (TBS) Tween 20 (TBST). For His tag detection, the protein expression of recombinant CHIKV proteins and viral antigens was evaluated at 1:5,000 and 1:10,000 dilutions. The horseradish peroxidase (HRP)-enhanced chemiluminescence (ECL) assay was used, and the color-developing solution was mixed with solution A and solution B at a 1:1 ratio before adding to the NC film. After placing the film into the black box, the film was colored and stored.

**CHIKV IgG ELISA kit.** A human chikungunya virus IgG antibody (CHIKV IgG) ELISA commercial kit (JC, China, JL21656) was purchased as a control to verify the accuracy of the LISA method. The experimental process was performed in strict accordance with the manufacturer's instructions. Briefly, the kit was placed at room temperature for 30 min before use. The samples in the kit were used as the positive control. The CHIKV E1 RT-PCR-positive patient serum samples were diluted with buffer at a ratio of 1:50 and incubated at 37°C for 60 min. The liquid was discarded, and the 96-well plate was patted dry with absorbent paper. Each hole was filled with TBST washing liquid and left standing for 1 min. The washing liquid was discarded, and the plate was patted dry with absorbent paper. The plate (Corning, USA, 3922) was washed five times. Substrate A (50 $\mu$L) and substrate B (50 $\mu$L) were added to each well and were incubated at 37°C for 15 min. Stop solution (50 $\mu$L) was added to each well. The optical density (OD) reading at 450 nm was taken using a multifunctional microplate instrument (Tecan, Spark 10M).

**CHIKV IgM ELISA kit.** Human chikungunya virus antibody IgM (CF IgM) ELISA commercial kits (Yi Bai Shun, China, JM-0609H2) were used to detect IgM in the sera of patients infected with CHIKV. All standards and samples were added in duplicate to the Micro-ELISA strip plate. First, the positive and negative controls (50 $\mu$L) were added separately to the respective wells. The test samples (10 $\mu$L) were then added to the sample wells, followed by sample diluent (40 $\mu$L). Next, 100 $\mu$L of HRP-conjugate reagent was added to each well, covered with an adhesive strip, and incubated for 60 min at 37°C. Each well was then aspirated and washed, repeating the process four times for a total of five washes. Next, the wells were washed by filling each well with wash solution (400 $\mu$L) using a squirt bottle, manifold dispenser, or autowasher, completely removing the liquid at each step. After the last wash, any remaining wash solution was removed by aspirating or decanting, and the plate was inverted and blotted against clean paper towels. Chromogen solution A (50 $\mu$L) and chromogen solution B (50 $\mu$L) were then added to each well, gently mixed, and incubated for 15 min at 37°C in the dark. Finally, a stop solution (50 $\mu$L) was

added to each well until the color changed from blue to yellow. The OD at 450 nm was read using a microtiter plate reader within 15 min.

**Development of LISA based on different E-luciferase fusion proteins.** Protein G (5 $\mu$g/mL, 50 $\mu$L/well; Genscript, China, Z02007) was diluted to 5 $\mu$g/mL with phosphate-buffered saline (PBS; 0.01 M, pH 7.4), added to the whiteboard of 96-well flat-bottomed luminometer plates (Corning Costar, USA, 3922) at 100 $\mu$L per well, and coated at 4°C overnight. First, 5% skim milk in PBS was used as a sealing solution. The coated board was then washed with 0.05% Tween in PBS (PBST) after a 1-h incubation of the sealing solution. Next, 300 $\mu$L of sealing solution was added to each well, and the wells were incubated at 37°C in an incubator for 1 h. Following incubation, the wells were washed with PBST three times and dried as much as possible after the final wash. Next, the sample was diluted with 2% skim milk (Sangon Biotech, China, A600669-0250) and added to the whiteboard at 100 $\mu$L per well. Positive, negative, and blank controls were set in each experiment. Next, the whiteboard was incubated at 37°C for 1 h. Following incubation, the wells were washed with PBST five times and dried as much as possible after the final wash. The supernatant of luciferase target antigen was then diluted with 2% skim milk (CHIKV E antigen diluted 1,000-fold). Next, 50 $\mu$L was added to each well and incubated in a 37°C incubator for 1 h. Following incubation, the wells were washed with PBST five times and dried as much as possible after the final wash. Finally, 50 $\mu$L of luciferase substrate (Promega, USA, N1120) was added per well according to the instructions. The fluorescence value was measured within 2 h using a fluorophotometer.

All steps were performed according to the manufacturer's instructions. We comprehensively analyzed the RFI value of healthy people and the CHIKV-infected population. To ensure specificity, we compared the mean + SEM in the CHIKV-infected population, which is approximately two times the RFI value of the control group, defined as anti-CHIKV IgG antibody positive. We set the anti-CHIKV IgG antibody-positive RFI value as 50,000. Data were expressed as the mean RFI of parallel duplicate wells and were corrected for background by subtracting the RFI value of the well incubated with 293T cell extracts in the absence of sera. Data were expressed as the mean RFI of parallel duplicate wells and were corrected for background by subtracting the RFI value of the well incubated with 293T cell extracts in the absence of sera.

To avoid differences in transfection efficiency and protein expression of different batches of preparations, we measured the luciferase activity of crude cell lysates to determine the RFI, which is usually between $10^4$ and $10^6$. The CHIKV antigen was always added in each reaction with $10^5$ RFI for LISA. In addition, we included positive and negative controls in each reaction plate to keep the results consistent and reproducible.

**Statistical analysis.** Data are presented as means $\pm$ standard deviation (SD) or means $\pm$ SEM. As stated in the figure labels, differences between groups and controls were analyzed using appropriate statistical tests. A $P$ value of $<0.05$ was considered significant. Statistical analyses were performed using GraphPad Prism 5.0 software and SPSS 20.

## ACKNOWLEDGMENTS

We thank Zhihua Liu of the Nan Fang Hospital, Southern Medical University, for the samples and related information. We thank the Key Laboratory of Liver Disease, Nan Fang Hospital, Southern Medical University, for providing samples of patients with hepatitis C (HCV).

This work was supported by the Natural Science Foundation of Guangdong Province (number 2018B030311063) and Guangdong Science and Technology Program Key Projects (number 2021B1212030014).

We declare no conflicts of interest.

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
