## [Reviewer comments · Microbiology Spectrum]

Microbiology Spectrum

Luciferase Immunosorbent Assay Based on Multiple E Antigens for the Detection of Chikungunya Virus-specific IgG Antibodies

Xiaoxia Li, Jinyue Liu, Haiying Wang, Anan Li, Changwen Ke, Shixing Tang, and Chengsong Wan

Corresponding Author(s): Chengsong Wan, Southern Medical University

Review Timeline:

Submission Date:	September 5, 2021
Editorial Decision:	October 28, 2021
Revision Received:	December 2, 2021
Editorial Decision:	December 21, 2021
Revision Received:	January 14, 2022
Accepted:	January 18, 2022

Editor: Manjula Kalia

Reviewer(s): Disclosure of reviewer identity is with reference to reviewer comments included in decision letter(s). The following individuals involved in review of your submission have agreed to reveal their identity: Ralph Martin Henri Gerard Huits (Reviewer #3)

Transaction Report:

DOI: <https://doi.org/10.1128/Spectrum.01496-21>

October 28, 2021

Dr. Chengsong Wan
Southern Medical University
Guangzhou
China

Re: Spectrum01496-21 (Luciferase Immunosorbent Assay Based On Multiple E Proteins for the Detection of Chikungunya Virus-Specific IgG)

Dear Dr. Chengsong Wan:

Thank you for submitting your manuscript to Microbiology Spectrum. Your manuscript has been reviewed by two experts. Both the reviewers have raised major concerns regarding the study, manuscript organisation and readability. In the current form the manuscript difficult to read and understand. A major revision for content and structure with professional language editing service is required for the manuscript to be considered further.

When submitting the revised version of your paper, please provide (1) point-by-point responses to the issues raised by the reviewers as file type "Response to Reviewers," not in your cover letter, and (2) a PDF file that indicates the changes from the original submission (by highlighting or underlining the changes) as file type "Marked Up Manuscript - For Review Only". Please use this link to submit your revised manuscript - we strongly recommend that you submit your paper within the next 60 days or reach out to me. Detailed information on submitting your revised paper are below.

Link Not Available

Sincerely,

Manjula Kalia

Journals Department
Reviewer comments:

Reviewer #2 (Comments for the Author):

The aim of the study is to establish a more sensitive and efficient diagnostic technique for chikungunya. But the need or importance of it did not come across in the manuscript. Data needs better interpretation and representation.

Please see the following comments:

1. Numbering of figures and tables is not correct. Figure 1 and table 1 has been labeled as figure 2 and table2; hence all the numbering is wrong going forward.
2. The constructs and blot in currently labeled fig 2 have the name of constructs as E1/2-CZ1/4 while in the text these have been mentioned as E1/2-C1/4 (line 124). Please choose one label throughout the manuscript.
3. The figure legends for current figure 4 are missing. Also, the graph needs to be aligned and labeled properly.
4. Please recheck the legends for figure 6.

5. Right-hand side Graph in figure 6 is not clearly visible. Fig 6B: Labels at the x-axis should be properly aligned with the bars in the graphs.
 6. Graph on page 11 has no numbers, labels, and legends.
 7. Line 154, 182: Please mention the group clearly whose mean+SEM has been taken as threshold.
 8. Line 156: What are partial positive samples?
 9. Line 172: Please mention the total number of samples here out of which 2-3 are positive.
 10. Line 179: CHIKV has been mislabeled as ZIKV.
 11. Line 182 is contradicting line 214. Please explain the data more clearly.
 12. Line 230: According to the dilution suggested above this line for LISA and in this line for ELISA, the sensitivity seems to be 100 fold higher and not 10 fold. Please recheck the calculations.
 13. Please analyze the data using some statistical tool; it will help in data interpretation and understanding.
 14. Increasing the sample size will help to improve the study.
 15. Please read the manuscript thoroughly and make sure the figures and text are incoherent as right now the manuscripts hold figures which are not discussed in the text at all.
- Minor suggestions:
1. If you can put the cloning strategy (fig. 2D) before the expression blot, it's easier for the reader to comprehend.

Reviewer #3 (Comments for the Author):

The manuscript Spectrum01496-21 entitled 'Luciferase Immunosorbent Assay Based On Multiple E Proteins for the Detection of Chikungunya Virus-Specific IgG' is the report of development and evaluation of a CHIKV-specific LISA, that contains important data that could further the development and commercialization of a novel technology. In its current form and structure, I find the work difficult to read - going back and forth between sections to look for information that may be provided, or not. I would prefer the traditional order of the Methods section preceding the Results.

To explain the main message of this work to a non-expert audience, a brief discourse on CHIKV/ alphavirus structure is required; the authors can elaborate on l.96 and following.

l.58 re-emergence 2006-2010, spread: indicate the spread of CHIKV genotypes as done in paragraph 2.

l.67 types: genotypes? Lineages?

l.79 ref. 12 cites an editorial by Wang that does not provide information on CHIKV-specific antibody kinetics. For a discussion on antibody/ RT-PCR kinetics see <https://doi.org/10.1371/journal.pone.0196630> or <https://doi.org/10.1093/infdis/jiw274>

l.83 alphaviruses or genus *Alphavirus* (italic).

l. 102 cited reference describes direct plasmid expression in insect cells as a means of generating recombinant protein for CHIKV diagnostics.

l.114 which CHIKV strains were used for plasmid expression? And which plasmids?

l.128 define and explain what is meant by 'correct'?

l. 169 and following:

a) for samples and controls appropriate annotation of these samples is required. What genotype CHIKV, which outbreak/ geographic/ time origin, handling and storage of the samples prior to testing. In control samples: how can we be certain they had no prior CHIKV exposure? Note: the information is partially provided in the Discussion and Table

b) when discussing 'better', please provide appropriate statistics of comparison in the main text.

l.179 ?ZIKV infected cases?

l.273 What are positive serum IgG antibodies.

l. 402 Commercial kits were purchased for use as a control- provide detail/ manufacturer

Minor: Explain abbreviations at first use (e.g. l. 454, standard error of the mean (SEM); relative fluorescence intensity (RFI), both used extensively in section 3)

Check for typos and have the manuscript checked by a native English speaker or a Language Editing Service. Examples: IMPROTANCE; to 'strengthen' CHIKV detecting ; 'recessive' infection; 'stronger' specificity.

Staff Comments:

Preparing Revision Guidelines

- Point-by-point responses to the issues raised by the reviewers in a file named "Response to Reviewers," NOT IN YOUR COVER LETTER.
- Upload a compare copy of the manuscript (without figures) as a "Marked-Up Manuscript" file.

- Each figure must be uploaded as a separate file, and any multipanel figures must be assembled into one file.
- Manuscript: A .DOC version of the revised manuscript
- Figures: Editable, high-resolution, individual figure files are required at revision, TIFF or EPS files are preferred

Please return the manuscript within 60 days; if you cannot complete the modification within this time period, please contact me. If you do not wish to modify the manuscript and prefer to submit it to another journal, please notify me of your decision immediately so that the manuscript may be formally withdrawn from consideration by Microbiology Spectrum.

Please see the following comments:

1. Numbering of figures and tables is not correct. Figure 1 and table 1 has been labeled as figure 2 and table2; hence all the numbering is wrong going forward.
2. The constructs and blot in currently labeled fig 2 have the name of constructs as E1/2-CZ1/4 while in the text these have been mentioned as E1/2-C1/4 (line 124). Please choose one label throughout the manuscript.
3. The figure legends for current figure 4 are missing. Also, the graph needs to be aligned and labeled properly.
4. Please recheck the legends for figure 6.
5. Right-hand side Graph in figure 6 is not clearly visible. Fig 6B: Labels at the x-axis should be properly aligned with the bars in the graphs.
6. Graph on page 11 has no numbers, labels, and legends.
7. Line 154, 182: Please mention the group clearly whose mean+SEM has been taken as threshold.
8. Line 156: What are partial positive samples?
9. Line 172: Please mention the total number of samples here out of which 2-3 are positive.
10. Line 179: CHIKV has been mislabeled as ZIKV.
11. Line 182 is contradicting line 214. Please explain the data more clearly.
12. Line 230: According to the dilution suggested above this line for LISA and in this line for ELISA, the sensitivity seems to be 100 fold higher and not 10 fold. Please recheck the calculations.
13. Please analyze the data using some statistical tool; it will help in data interpretation and understanding.
14. Increasing the sample size will help to improve the study.
15. Please read the manuscript thoroughly and make sure the figures and text are incoherent as right now the manuscripts hold figures which are not discussed in the text at all.

Minor suggestions:

1. If you can put the cloning strategy (fig. 2D) before the expression blot, it's easier for the reader to comprehend.

Reviewer #2:

1. Numbering of figures and tables is not correct. Figure 1 and table 1 has been labeled as figure 2 and table2; hence all the numbering is wrong going forward.

response : I am sorry to confuse the order of the charts, we changed FIG2 to FIG1.

FIG 1 The figure and table numbers have been revised, the labeling in figure 1 and table 1 have been revised, and all the numbers have been corrected throughout the manuscript.

2. The constructs and blot in currently labeled fig 2 have the name of constructs as E1/2-CZ1/4 while in the text these have been mentioned as E1/2-C1/4 (line 124). Please choose one label throughout the manuscript.

response: Line 129-132. In Figure 1B and Figure 1D, including full-length E1, E1-C1, E1-C2, E2 full-length, E2-C1, E2-C2, E2-C3, and E2-C4, all of which were transfected into 293T cells to express the luciferase fusion expressed protein (Fig.1B, Fig.1D). Line 129-132, the blot in currently labeled FIG 2 have the name of constructs as E1/2-C1/4, We have chosen one label throughout the manuscript.

3. The figure legends for current figure 4 are missing. Also, the graph needs to be aligned and labeled properly.

response: We have modified FIG4 to FIG3, as shown in FIG 3A and FIG 3B have been added to the legend and labeled.

Figure 3B: statistical analysis of comparison between CHIKV-E1 antigen and CHIKV-E2 antigen. The Wilcoxon rank sum test was used to analyze the detection conditions based on CHIKV-E1 and CHIKV-E2, and the *P* of the two-tailed data test was 0.024 ($P < 0.05$), which was statistically significant. (Fig. 3B)

4. Please recheck the legends for figure 6.

response: Thank you for pointing out the error. We changed FIG6 to FIG5. The legends for FIG 5 have rechecked, we rejoin the legend.

5. Right-hand side Graph in figure 6 is not clearly visible. Fig 6B: Labels at the x-axis should be properly aligned with the bars in the graphs.

response: We reworked the drawing, and we replace FIG 6 with FIG5. FIG 5B has

been changed to an analysis statistics chart due to unclear expression. We use ELISA-E1 and LISA-E2 two detection assays to detect the number of CHIKV IgG antibody samples and two sample rate chi-square test system statistical interpretations, the P of the two-tailed data test was 0.004 ($P < 0.05$), which was statistically significant (Fig.5B)

6. Graph on page 11 has no numbers, labels, and legends.

response : I'm very sorry, there were some problems during the production of Graph on page 11. This picture does not really express what we want. We changed graph on page 11 to TABLE 3, which is the detection rate and cross-reactivity of different detection assays.

7. Line 154, 182: Please mention the group clearly whose mean+ SEM has been taken as threshold.

response : Line150-160 and In TABLE 1, we have added a summary of sample information, please ref TABLE1. According to the patient's number in TABLE, we comprehensively analyze the relative fluorescence intensity (RFI) value of normal people and CHIKV-infected population. To ensure specificity, we use compare the mean + standard error of the mean (SEM) in the CHIKV-infected population which approximately 2 times the RFI value of the control group. We defined anti-CHIKV IgG antibody positive RFI cut-off value as 5×10^4 , and consider it a positive result when CHIKV-infected patients were tested with the LISA assay and the RFI value is higher than 5×10^4 . We also summarize the number of CHIKV IgG antibody-positive patients.

8. Line 156: What are partial positive samples?

response : Please refer to Line 163-165. We reinterpreted the "partial positive samples": The constructed LISA detection method based on CHIKV-E1 protein and CHIKV-E2 protein can detect serum CHIKV IgG antibodies in some patients with CHIKV infection (FIG.2).

9. Line 172: Please mention the total number of samples here out of which 2-3 are

positive.

response : In TABLE1, we summarized part of the information about CHIKV-infected patients (some of which is missing), please refer to us for mentioning the total number of patients.

Line 166-170 The results of the recombinant Nano-luciferase CHIKV-E protein antigen in the serum of patients showed that the recombinant CHIKV-E1 protein could only detect CHIKV IgG antibodies in 2-3 out of 20 CHIKV infected samples, which were similar with the results of existing commercial kits. CHIKV-E2 recombinant protein could detect more than half of patients' samples.

10. Line 179: CHIKV has been mislabeled as ZIKV.

response : We are very sorry, but we misused CHIKV and ZIKV. We have made corrections , and we reviewed the full text for any misuse.

11. Line 182 is contradicting line 214. Please explain the data more clearly.

response: Line 201-202 As a result, not all the IgG antibody in the sample can bind to the CHIKV E2-full antigen. What we want to state is that CHIKV-E2 whole antigen formation does not bind to all CHIKV IgG antibody positive samples. Because E2 may form a space folded structure, some important binding points in the RBD region of CHIKV are hidden inside the E2 antigen.

12. Line 230: According to the dilution suggested above this line for LISA and in this line for ELISA, the sensitivity seems to be 100 fold higher and not 10 fold. Please recheck the calculations.

response: Thank you very much for your question. We found the original data and re-analyzed it. Some problems have arisen due to unclear data presentation. The results of our experiment are as follows: The ELISA for detection of CHIKV-infected IgG antibody-positive serum dilution range is 50-200 times, and the sample dilution range for LISA detection is 1000-5000 times. After about 5000 times dilution, CHIKV IgG positive samples will not be detected, and the sensitivity of the LISA was about 25-fold higher than that of ELISA.

13. Please analyze the data using some statistical tool; it will help in data interpretation and understanding.

response: After listening to your suggestions, FIG 5B and FIG3A statistical analysis data are added.

Due to the limitation of sample size, the sample data does not conform to the normal distribution. The Wilcoxon rank sum test was used to analyze the detection conditions based on CHIKV-E1 and CHIKV-E2, and the P of the two-tailed data test was 0.024 ($P < 0.05$), which was statistically significant (Fig. 3B).

We use ELISA-E1 and LISA-E2 two detection methods to detect the number of CHIKV IgG antibody samples and two sample rate chi-square test system statistical interpretations, the P of the two-tailed data test was 0.004 ($P < 0.05$), which was statistically significant (Fig.5B).

14. Increasing the sample size will help to improve the study.

response: Thank you very much for your suggestions. Actually, the samples we got are limited. Chikungunya virus rarely occurs in China. We collected all existing samples in Guangdong Province for analysis. The existing samples can represent the occurrence of Chikungunya virus in Guangdong Province.

15. Please read the manuscript thoroughly and make sure the figures and text are incoherent as right now the manuscripts hold figures which are not discussed in the text at all.

response: We have read the manuscript carefully again to ensure that the graphics and text are coherent. Since the author's mother tongue is not English, it does increase the difficulty of reading. We will ask someone to help modify the language, and ensure the readability of the article.

Minor suggestions:

1. If you can put the cloning strategy (fig. 2D) before the expression blot, it's easier for the reader to comprehend.

response : Thank you for your suggestion, FIG 1 we have putted the cloning strategy before the expression blot.

Reviewer #3

To explain the main message of this work to a non-expert audience, a brief discourse on CHIKV/ alphavirus structure is required; the authors can elaborate on 1.96 and following.

response: Ref line 111-121, We re-added a brief introduction to CHIKV and the structure of CHIKV (including genome size and structure, and functions of related proteins). A brief discourse on CHIKV/ alphavirus structure has supplied (FIG 1A).

1.58 re-emergence 2006-2010, spread: indicate the spread of CHIKV genotypes as done in paragraph 2.

response: Ref line51-55 Since its re-emergence in epidemic proportions in several southeast Asian countries which the epidemic strains of CHIKV are Asian and IOL types during 2006–2010(2,3). We collected sample information (TABLE 1), introduced the processing methods and storage conditions of serum samples.

1.67 types: genotypes? Lineages?

response: Ref Line 62 the current types include the Asian type, East/Central Africa/South Africa type (ECSA), West African type, and the Indian Ocean Lineage type (IOL).

1.79 ref. 12 cites an editorial by Wang that does not provide information on CHIKV-specific antibody kinetics. For a discussion on antibody/ RT-PCR kinetics see <https://doi.org/10.1371/journal.p one.0196630> or <https://doi.org/10.1093/infdis/jiw274>

response: ref. (12) cites have changed to Yue C., Teitz S., Miyabashi T., Boller K., Lewis-Ximenez L.L., Baylis S.A., Blümel J. 2019. Inactivation and removal of chikungunya virus and mayaro virus from plasma-derived medicinal products. *Viruses* 11:234. At the same time, we verified all cited documents. The reference 4, 12, 13 and 19 documents have been replaced.

1.83 alphaviruses or genus *Alphavirus* (italic).

response: Thank you for your suggestion, we have corrected it. Line78 has marked

Alphavirus in italics.

l. 102 cited reference describes direct plasmid expression in insect cells as a means of generating recombinant protein for CHIKV diagnostics.

response: Due to inappropriate citations in the article, we changed the literature in the article. Please refer to Ref number 19.

Young Chan Kim, César López-Camacho, Arturo Reyes-Sandoval. Development of an E2 ELISA Methodology to Assess Chikungunya Seroprevalence in Patients from an Endemic Region of Mexico[J]. *Viruses*, 2019, 11(5):407.

l.114 which CHIKV strains were used for plasmid expression? And which plasmids?

response: Line 374-385 The recombinant antigen sequence of CHIKV rE2 (strain S27-African prototype) was 1275 nucleotides (nt) in length, without the transmembrane region (Genbank accession number, AF369024). Standard molecular cloning was performed to clone E1 glycoprotein (rE1: amino acids 1–499) and E2 glycoprotein (rE2: amino acids 1–425) into a pNLF1-N (Promega, USA) vector, which contains luciferase expression gene (Fig.1B).

l.128 define and explain what is meant by 'correct'?

response: Line 133-139 These plasmids were confirmed by restriction endonuclease reactions and gel electrophoresis. Recombinant protein was detected using the mouse monoclonal antibody against CHIKV E and His tags fusion expression protein by western blot (Fig.1D). The results of sequence comparison after gene sequencing and western blot confirmed the correct construction and expression of the CHIKV E plasmid. LISA based on multiple E antigens can be applied to the detection of CHIKV-specific IgG (Fig. 1 E).

l. 169 and following:

a) for samples and controls appropriate annotation of these samples is required. What genotype CHIKV, which outbreak/ geographic/ time origin, handling and storage of the samples prior to testing. In control samples: how can we be certain they had no prior CHIKV exposure? Note: the information is partially provided in the Discussion and Table.

b) when discussing 'better', please provide appropriate statistics of comparison in the

main text.

response:

- a) Line 334-339 Genetic analysis of the 325-nt fragment of E1 genes obtained in this study showed that all 7 sequences clustered in a unique branch within the Indian Ocean clade of the East/Central/South African (ECSA) genotype, and close to Thailand (GQ870312, FJ882911, GU301781), Malaysia (FJ998173), Taiwan (FJ807895), and China (GU199352, GU199353) isolates (98%–99%). refer to: TABLE 1 Summary of sample detection in this study.
- b) Line 181-185 Due to the limitation of sample size, the sample data does not conform to the normal distribution. The Wilcoxon rank sum test was used to analyze the detection conditions based on CHIKV-E1 and CHIKV-E2, and the P of the two-tailed data test was 0.024 ($P < 0.05$), which was statistically significant (Fig. 3B)

1.179? ZIKV infected cases?

response: We are very sorry, but we misused CHIKV and ZIKV. We have made corrections , and we reviewed the full text for any misuse.

1.273 What are positive serum IgG antibodies.

response : Line 150-156 We comprehensively analyze the relative fluorescence intensity (RFI) value of normal people and CHIKV-infected population. To ensure specificity, we use compare the mean + standard error of the mean (SEM) in the CHIKV-infected population which approximately 2 times the RFI value of the control group. We defined anti-CHIKV IgG antibody positive RFI cut-off value as 5×10^4 , and consider it a positive result when CHIKV-infected patients were tested with the LISA assay and the RFI value is higher than 5×10^4 .

1. 402 Commercial kits were purchased for use as a control- provide detail/manufacturer

response : We have supplemented the information about commercial ELISA kits in the article.

Line 395 Human Chikungunya virus IgG antibody (CHIKV-IgG) ELISA commercial

kits (JC, China).

Line 409 Human Chikungunya Antibody IgM (CF IgM) ELISA commercial kits (Yi Bai Shun, China)

Minor: Explain abbreviations at first use (e.g. l. 454, standard error of the mean (SEM); relative fluorescence intensity (RFI), both used extensively in section 3)

response : Thank you for your questions and suggestions. We have revised the questions you mentioned in the full text.

Check for typos and have the manuscript checked by a native English speaker or a Language Editing Service. Examples: IMPROTANCE; to 'strengthen' CHIKV detecting; 'recessive' infection; 'stronger' specificity.

response : We have read the manuscript carefully again to ensure that the graphics and text are coherent. And ask someone to help modify the language.

December 21, 2021

Dr. Chengsong Wan
Southern Medical University
Guangzhou
China

Re: Spectrum01496-21R1 (Luciferase Immunosorbent Assay Based On Multiple E Proteins for the Detection of Chikungunya Virus-Specific IgG)

Dear Dr. Chengsong Wan:

Thank you for submitting your manuscript to Microbiology Spectrum. While your manuscript has improved, one reviewer has raised some important points which need to be addressed.

Further please address the following issues:

1. While the readability of the manuscript has considerably improved, it still has several spelling and grammatical mistakes (More than 10). I strongly recommend professional language editing services. Please see the following link: <https://journals.asm.org/language-editing-services>
2. The figure legends for Fig 2, 3 and 5 are very brief and should be elaborated.
3. For several of the commercial kits and other reagents the catalogue numbers should be specified.
4. Does the MS have supplemental material as mentioned in line 471?

Link Not Available

Sincerely,

Manjula Kalia

Journals Department
Reviewer comments:

Reviewer #2 (Comments for the Author):

The study has been improved. Data is more refined and well discussed. A few minor concerns need to be addressed:

1. Figure 1: The names of the clones in 1B & 1D are different. E.g., E1-C1 (1D) and C1- E1(1B).
2. Figure 2A: The graph for E1-C1 has three samples over the cut-off but only two have been mentioned in the text.
3. Figure 2B: Two graphs have been labeled as E2-C2 while E2-C3 is missing.

4. Figure 3B: The lines drawn on the graph failed to depict which two groups have been compared.
5. Figure 4: Title C is not aligned.
6. Line 183: Can the author explain why E2-C2 has not been mentioned here even though it has a higher positivity rate than E2-C1 & E2-C3.

Staff Comments:

Preparing Revision Guidelines

Please return the manuscript within 60 days; if you cannot complete the modification within this time period, please contact me. If you do not wish to modify the manuscript and prefer to submit it to another journal, please notify me of your decision immediately so that the manuscript may be formally withdrawn from consideration by Microbiology Spectrum.

The study has been improved. Data is more refined and well discussed. A few minor concerns need to be addressed:

1. Figure 1: The names of the clones in 1B & 1D are different. E.g., E1-C1 (1D) and C1- E1 (1B).
2. Figure 2A: The graph for E1-C1 has three samples over the cut-off but only two have been mentioned in the text.
3. Figure 2B: Two graphs have been labeled as E2-C2 while E2-C3 is missing.
4. Figure 3B: The lines drawn on the graph failed to depict which two groups have been compared.
5. Figure 4: Title C is not aligned.
6. Line 183: Can the author explain why E2-C2 has not been mentioned here even though it has a higher positivity rate than E2-C1 & E2-C3.

Response to Editors and Reviewers :

Editor comments:

1. While the readability of the manuscript has considerably improved, it still has several spelling and grammatical mistakes (More than 10). I strongly recommend professional language editing services. Please see the following link: <https://journals.asm.org/language-editing-services>

response : Thank you for your suggestion. Based on your recommendation, we edited the language again. A certificate of English language editing is attached.

2. The figure legends for Fig 2, 3 and 5 are very brief and should be elaborated.

response : We have reworked the pictures, and the legends for Fig 2, 3 and 5 have been further explained.

3. For several of the commercial kits and other reagents the catalogue numbers should be specified.

response : Thank you very much for your suggestions. We have added the commercial kits and other agents the catalog numbers.

4. Does the MS have supplemental material as mentioned in line 471?

response : Due to wrong understanding , we are very sorry that the supplementary material mentioned in line 489 does not. All our information is provided in the article.

Reviewer comments:

Reviewer #2 (Comments for the Author):

The study has been improved. Data is more refined and well discussed. A few minor concerns need to be addressed:

1. Figure 1: The names of the clones in 1B & 1D are different. E.g., E1-C1 (1D) and C1-E1(1B).

response : Thank you very much for your suggestion. We haven't noticed this problem. As shown in Figure 2A, we have modified it according to your suggestion.

2. Figure 2A: The graph for E1-C1 has three samples over the cut-off but only two have been mentioned in the text.

response : Thank you very much for your advice. As you said, three samples were detected in Table 2. We did not mention one sample because of our mistake. As shown in Figure 2A, the mistake has been corrected.

3. Figure 2B: Two graphs have been labeled as E2-C2 while E2-C3 is missing.

response : Thank you very much for pointing out the mistakes. As shown in Figure 2B , we have corrected the mistakes you pointed out.

4. Figure 3B: The lines drawn on the graph failed to depict which two groups have been compared.

response : As shown in Figure 3B, the comparisons between two groups were marked.

5. Figure 4: Title C is not aligned.

response : As shown in Figure 4, Title 4 has been aligned.

6. Line 183: Can the author explain why E2-C2 has not been mentioned here even though it has a higher positivity rate than E2-C1 & E2-C3.

response : Line 195-197, We discussed E2-C2: E2-C2 has a higher positivity rate than E2-C1 and E2-C3. This also suggests that there are also CHIKV IgG antibody binding sites in the Domain A and Domain B. However, we believe that Domain A has a more significant effect than Domain B.

January 18, 2022

Dr. Chengsong Wan
Southern Medical University
Guangzhou
China

Re: Spectrum01496-21R2 (Luciferase Immunosorbent Assay Based on Multiple E Antigens for the Detection of Chikungunya Virus-specific IgG Antibodies)

Dear Dr. Chengsong Wan:

Your manuscript has been accepted, and I am forwarding it to the ASM Journals Department for publication. You will be notified when your proofs are ready to be viewed.

Sincerely,

Manjula Kalia
Editor, Microbiology Spectrum
